# HIV-1 DNA sequence diversity and evolution during acute subtype C infection

Guinevere Q. Lee [1,2,3], Kavidha Reddy [4,5], Kevin B. Einkauf [1,2], Kamini Gounder[4,5], Joshua M. Chevalier [1], Krista L. Dong[1,3], Bruce D. Walker [1,2,3,4,6,7], Xu G. Yu[1,2,3,6], Thumbi Ndung'u [1,4,5,8] & Mathias Lichterfeld[1,2,3,6]

Little is known about the genotypic make-up of HIV-1 DNA genomes during the earliest stages of HIV-1 infection. Here, we use near-full-length, single genome next-generation sequencing to longitudinally genotype and quantify subtype C HIV-1 DNA in four women identified during acute HIV-1 infection in Durban, South Africa, through twice-weekly screening of high-risk participants. In contrast to chronically HIV-1-infected patients, we found that at the earliest phases of infection in these four participants, the majority of viral DNA genomes are intact, lack APOBEC-3G/F-associated hypermutations, have limited genome truncations, and over one year show little indication of cytotoxic T cell-driven immune selections. Viral sequence divergence during acute infection is predominantly fueled by single-base substitutions and is limited by treatment initiation during the earliest stages of disease. Our observations provide rare longitudinal insights of HIV-1 DNA sequence profiles during the first year of infection to inform future HIV cure research.

[1] Ragon Institute of MGH, MIT and Harvard, Cambridge, MA 02139, USA. [2] Brigham and Women's Hospital, Boston 02115 MA, USA. [3] Harvard Medical School, Boston, MA 02115, USA. [4] HIV Pathogenesis Programme, Nelson R. Mandela School of Medicine, University of KwaZulu-Natal, Durban 4001, South Africa. [5] Africa Health Research Institute, Durban 4001, South Africa. [6] Broad Institute of MIT and Harvard, Cambridge, MA 02142, USA. [7] Howard Hughes Medical Institute, Chevy Chase, MD 20815, USA. [8] Max Planck Institute for Infection Biology, 10117 Berlin, Germany. Correspondence and requests for materials should be addressed to M.L. (email: MLICHTERFELD@mgh.harvard.edu)

Current HIV-1 antiretroviral treatment successfully controls viral replication and has transformed HIV-infection from a fatal illness to a manageable chronic condition[1]. However, despite suppression of viral replication during treatment, studies have shown that pools of latent reservoirs persist long-term and may fuel viral rebound when antiviral suppression treatment is interrupted[2–4]. These reservoirs are extremely durable, not susceptible to therapeutic effects of currently available antiretroviral agents and have been refractory to recent experimental treatment approaches[5].

Recent results suggest that such viral reservoirs are established extremely early in the infection process[4–8], possibly even prior to detection of viremia in blood, and indicated that treatment initiation in humans during the earliest stages of infection does not prevent viral reservoir seeding[8]. Nevertheless, multiple studies have demonstrated that early treatment can serve to limit reservoir sizes and diversity[9–12], preserve immune functions[10,13–15], and in some cases was associated with virologic control when treatment was interrupted[16,17]. A common goal of current HIV-1 cure research is to understand the complexity and evolutionary dynamics of viral reservoir cells, and to develop interventional strategies that destabilize or reduce their long-term persistence[18–20].

Thus far, studies to evaluate longitudinal decay dynamics of early HIV-1 reservoirs have relied on PCR-based techniques such as quantitative PCR (qPCR) and/or droplet digital PCR (ddPCR) which both capture short portions of HIV-1 DNA genomes. These methods are limited by the fact that over 90% of the viral DNA genomes in long-term treated chronically-infected patients are defective and replication-incompetent, and are more likely to represent fossils of the replicative history of HIV-1 in a given patient, rather than a functionally-relevant viral reservoir able to fuel rebound viremia[21]. A selective assessment of genome-intact HIV-1 proviruses would be highly informative for understanding mechanisms of reservoir establishment, but require complex molecular analysis techniques ideally based on near full-genome sequencing of individual proviral species. A pioneering study by Bruner et al. analyzed peripheral blood mononuclear cells (PBMCs) in nine patients who initiated antiretroviral treatment early during the disease course (17–97 days after presumed date of infection) and found that only 2–10% of proviruses had intact genomes, thus inferring that defective proviruses rapidly accumulate during acute HIV-1 infections[7]. Yet, these studies were conducted with cell samples collected 10–145 months post-treatment initiation, and therefore may not reflect viral DNA compositions during acute HIV-1 infection but might instead be influenced by the decay kinetics of HIV-1 DNA sequences following treatment initiation. For this and other reasons, the structure and composition of intact HIV-1 DNA sequences detectable within days after HIV-1 transmission remain unknown and require further investigation.

In this report, we provide a longitudinal evaluation of HIV-1 DNA sequences in four subtype C HIV-1-infected individuals identified at stages II and V of acute infection (according to the classification system of Fiebig et al.[22]). These four patients are enrolled in the FRESH (Females Rising through Education, Support, and Health) cohort[23,24] in Durban, South Africa, in which young women at high risk for acquiring HIV-1 infection are encouraged to undergo HIV-1 screening tests twice a week. If HIV-1 infection is documented, these individuals are offered to start antiretroviral treatment immediately, frequently within hours after positive screening test results, and are invited to participate in an observational cohort study focusing on longitudinal evaluations of clinical, virological and immunological parameters. This study design has enabled us to identify individuals within days after HIV-1 transmission, and to capture HIV-1 DNA sequence profiles detectable at these extremely early stages of infection, both in individuals who opted for immediate treatment initiation and in persons who remained off-treatment. Given that initiation of antiretroviral treatment during such exquisitely early stages of viral infection may offer a unique window of opportunity for future clinical interventions focusing on HIV-1 cure and eradication, we here conduct in-depth analyses of the structure, composition, and longitudinal evolution of proviral HIV-1 DNA sequences in four FRESH cohort participants. We observe that HIV-1 DNA genomes detected at the earliest phase of infection, namely at Fiebig stage II (before treatment initiation), consist predominantly of genome-intact viral DNA and have limited genome truncations; APOBEC-3G/F-associated hypermutations are not observed in the limited sampling depth. Furthermore, early HIV-1 DNA sequence divergence at both Fiebig stage II and V is predominantly fueled by single-base substitutions. Over 1 year of longitudinal follow-up in two persistently viremic patients who remained untreated, genome-intact viral DNA genomes show little indication of cytotoxic T cell (CTL)-driven immune selections despite the presence of HLA-alleles that have been associated with immune-mediated viral escape mutations[25,26].

## Results

**Patient characteristics**. Four female individuals (Fig. 1 and Supplementary Fig. 1) identified in primary HIV-1 infection were non-randomly selected for this study to each represent a unique clinical course based on their baseline acute infection stages and their subsequent treatment status. All four patients (Pt) were selected for having protective human leukocyte antigen (HLA) genotypes (Table 1) to shed light on the dynamics of CTL-driven viral evolution over time and were longitudinally followed for approximately 1 year after infection. Pt 1 and Pt 2 were both captured at stage II of acute HIV-1 infection (Supplementary Table 1): Pt 1 immediately initiated antiretroviral therapy (tenofovir, emtricitabine, efavirenz, raltegravir) after first detection of HIV-1 plasma viremia, with raltegravir withdrawn 90 days after viral suppression, whereas Pt 2 remained untreated according to the treatment guidelines at the time of diagnosis for the entire follow-up period of 328 days. In contrast, Pt 3 and Pt 4 were both captured at stage V of acute infection (Supplementary Table 1): Pt 3 immediately received antiretroviral treatment (tenofovir, emtricitabine, efavirenz) when plasma HIV-1 RNA was first detectable, whereas Pt 4 remained untreated for the entire follow-up period of 332 days. Table 1 summarizes HLA class I genotypes and peak/maximum viremia levels of each patient. Each of the four patients had one protective HLA-B allele[26]; only Pt 1 had an HLA-B allele associated with rapid disease progression[26]. All four patients had detectable viremia at one point during the study: In Pt 1, Pt 2, and P4, viremia peaked at $10^5$–$10^7$ copies/ml, whereas Pt 3 was likely detected post-peak viremia at 42 days after the most recent negative HIV-1 PCR-test, and our record showed a maximum at 150 copies of HIV-1 RNA/ml at this time. Referring to Fig. 1, both patients who received antiretroviral therapy (Pt 1 and Pt 3) achieved virologic suppression at <46 days post-detection, whereas both untreated patients (Pt 2 and Pt 4) demonstrated a spontaneous graduate 2-log decrease of plasma viral load over 1 year, likely reflecting natural restriction of HIV-1 replication via innate and adaptive immune mechanisms. CD4 counts in all patients over the entire study duration were consistently around 1000 cells/μL.

**Intact HIV-1 DNA genomes predominate in acute infections**. In a cross-sectional analysis using PBMC collected at the earliest available timepoints, we noted that HIV-1 DNA copies/million

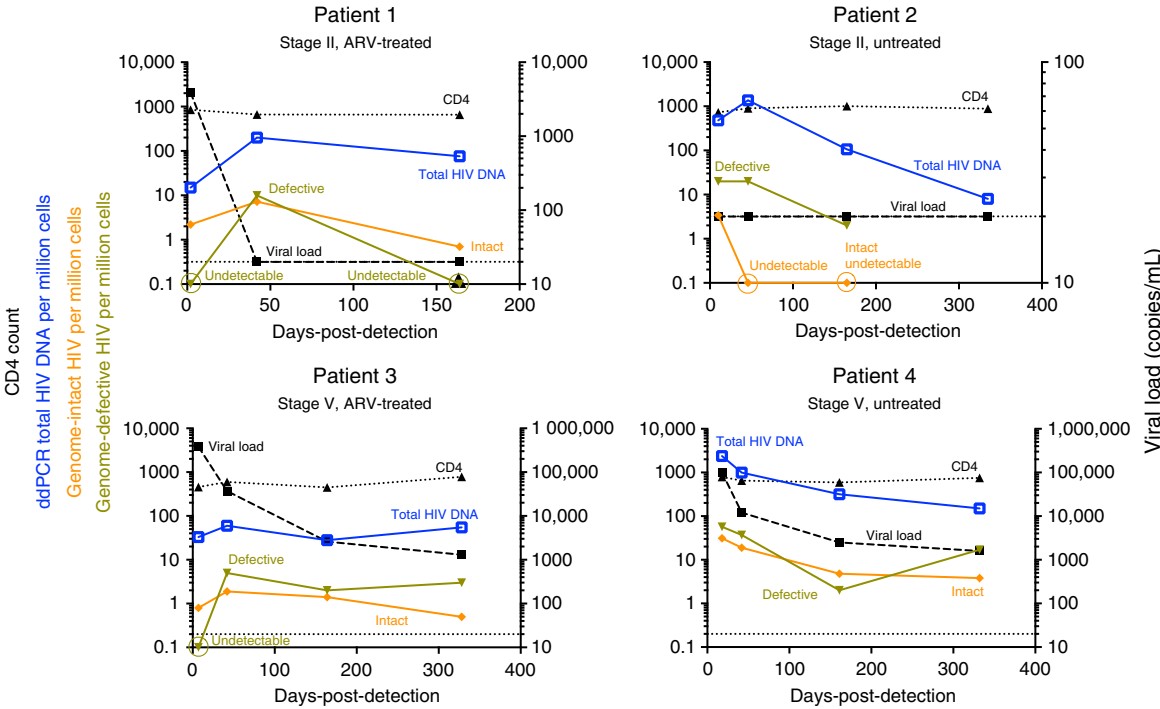

**Fig. 1** Clinical and virological characteristics of the four study participants. Longitudinal trends of viral load (dashed black lines), CD4 counts (dotted black lines), absolute frequencies of genome-intact HIV-1 per million PBMCs (orange lines), absolute frequencies of genome-defective HIV-1 (light green lines), and total HIV-1 DNA burden determined by ddPCR (blue lines) are shown. Acute HIV-1 infection was staged according to the classification system of Fiebig et al.[22]

### Table 1 Clinical and immunogenetic characteristics of study patients

| | FRESH ID | HLA-A | HLA-A | HLA-B | HLA-B | HLA-C | HLA-C | Stage of acute HIV-1 infection at time of earliest sample collection | Peak/maximum viremia (copies/ml) |
|---|---|---|---|---|---|---|---|---|---|
| Patient 1 | 127-33-0897-651 | 03:01 | 29:02 | 44:03[a] | 58:02[b] | 06:02 | 07:01 | Stage II | $2.6 \times 10^6$ |
| Patient 2 | 127-33-0397-268 | 01:01 | 66:01 | 39:10 | 81:01[a] | 12:03 | 18 | Stage II | $7.7 \times 10^5$ |
| Patient 3 | 127-33-0611-442 | 68:02 | 74 | 15:03 | 57:02[a] | 02:10 | 18 | Stage V | 150 |
| Patient 4 | 127-33-0262-198 | 23:01 | 30:01 | 15:10 | 58:01[a] | 03:02 | 16:01 | Stage V | $5.7 \times 10^7$ |

[a]HLA I alleles associated with protection[1]
[b]HLA I allele associated with disease-susceptibility[1]

PBMC determined by ddPCR were substantially lower in the two individuals identified during stage II of acute infection (Pt 1 and Pt 2: 15 and 33 copies/million PBMC), relative to the two remaining study persons diagnosed during stage V (Pt 3 and Pt 4: 480 and 2379 copies/million PBMC) (Fig. 1, earliest time point). However, this approach for viral DNA quantification relies on amplification of a short segment of the HIV-1 genome and does not allow to accurately quantify the frequency of intact proviral sequences which may evolve into functionally-relevant components of the long-term HIV-1 reservoir. In fact, the majority of viral sequences identified by ddPCR represent genome-defective HIV-1 DNA products that result from the high error rate of the viral reverse transcriptase and account for a considerable proportion of HIV-1 DNA sequences detectable in individuals undergoing antiretroviral therapy[7,21]. To address this, we performed single-genome, near-full-length next-generation HIV-1 DNA sequencing, followed by a complex biocomputational analysis procedure (Fig. 2), to specifically quantify relative proportions of genome-intact HIV-1 DNA sequences and viral DNA species exhibiting genome defects precluding viral replication. At each patient's earliest sampling time point available, we sampled

4.1, 1.3, 1.8, and 0.8 million PBMCs from Pt 1 to Pt 4, respectively, and detected a total of 42 genome-intact HIV-1 DNA sequences at the earliest sampling time points. Consistent with trends observed via ddPCR, absolute frequencies of genome-intact HIV-1 DNA in patients identified during stage II were lower than those at stage V (2.2 and 0.8 versus 3.4 and 31.1 genome-intact HIV-1 DNA copies per million PBMC, for patients 1–4, respectively; Fig. 3a). Yet, relative contributions of intact genomes to the total pool of HIV-1 DNA sequences were markedly higher in stage II compared to stage V (82%, 100% in Pt 1 and 2, versus 15%, 35% in Pt 3 and 4; Fig. 3b). For comparative purposes, we performed single-genome, near full-length HIV-1 DNA sequencing using PBMCs from three chronically-infected patients who had uncontrolled viremia for >4, >7, and >12 years of prior to initiation of ART and who underwent cell sampling at 18, 70, and 182 days after treatment onset. Notably, relative proportions of intact HIV-1 sequences among all HIV-1 DNA amplification products were substantially smaller in these three patients in comparison to all four patients identified in primary HIV-1 infection (Pt 5: 1% (1/175), Pt 6: 5% (3/64), and Pt 7: 2% (1/47)) (Fig. 3c). In these three patients, absolute frequencies of

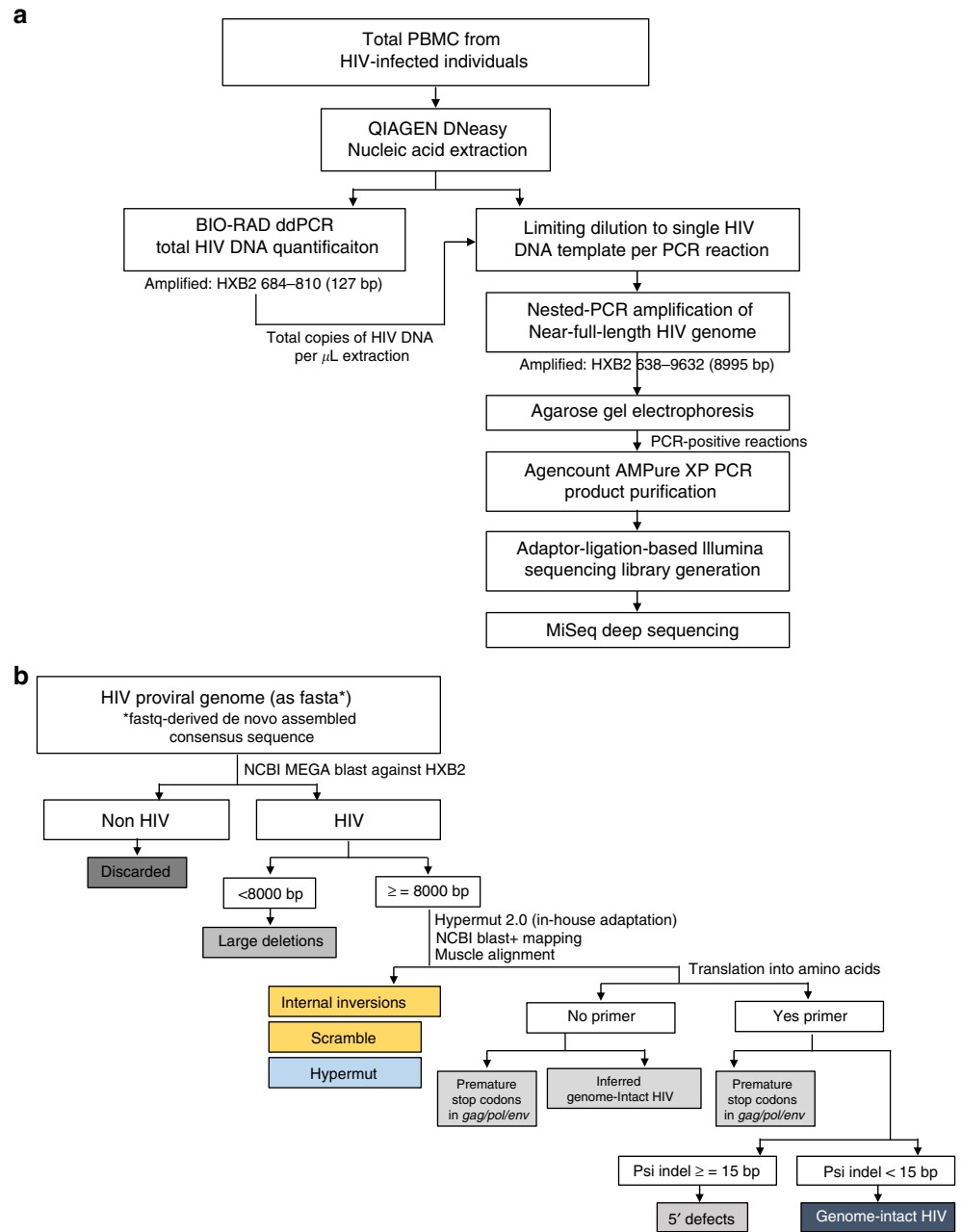

**Fig. 2** Experimental and computational determination of viral genome intactness. **a** Sample processing pipeline for the generation of near-full-length HIV-1 sequences via single-genome-amplification. **b** Schematic representation of individual viral sequence analysis steps incorporated in the R-language HIVSeqinR script used for the determination of viral genome intactness in this study. A stable release (ver2.6) used in this study is available in GitHub at https://github.com/guineverelee/HIVSeqinR

total HIV-1 DNA were 48.9, 39.3, and 35.9 copies per million PBMC, and absolute frequencies of genome-intact HIV-1 DNA were 0.3, 1.8, and 0.8 copies per million PBMC respectively (Supplementary Table 2). Together, these results suggest a predominance of genome-intact HIV-1 DNA sequences during early stages of HIV-1 infection.

**Spectrum of defective HIV-1 DNA genomes in acute infections.** We subsequently conducted a detailed analysis of the defective viral sequences observed in the four study patients with primary HIV-1 infection. Among the defective genomes detected, no APOBEC-3G/3F-hypermutated HIV-1 DNA genomes were observed at stage II after sampling 1.3–4.1 million PBMCs,

in contrast to 6% (2/35) and 6% (3/48) of hypermutated sequences detected at stage V after sampling 0.8–1.8 million PBMCs (Fig. 3d). This observation is limited by the small number of total HIV-1 DNA genomes detected at stage II. Non-hypermutated varieties of defective HIV-1 DNA genomes detected at stage II included: One case of double premature stop codons at amino acid positions 628 and 670 in Env (HXB2 coordinates), both attributable to TGG to TAG single-base substitution mutations (Pt 1), and one case of a single large internal deletion corresponding to HXB2 coordinate 7016–8912 spanning the V3 loop in gp120 to *nef* (Pt 1). No defective viral genomes were detected at Pt 2's initial analysis timepoint during stage II of acute infection (Supplementary Table 3, rows 1 and 4). In contrast, at stage V, non-hypermutated

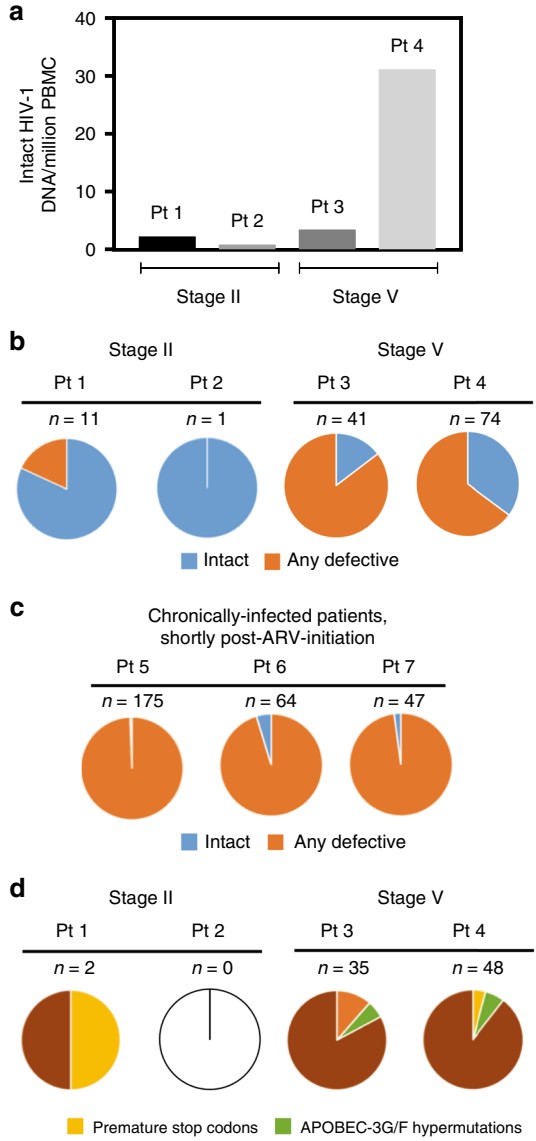

**Fig. 3** Cross-sectional analysis of HIV-1 DNA sequences at stage II versus stage V of acute infections. **a** Absolute frequencies of intact viral genomes detected per million PBMC in indicated study patients at the earliest sampling time points. **b** Proportions of intact (blue) and defective (orange) viral genomes at the earliest sampling time points available from each of the four FRESH cohort participants. **c** Proportions of intact (blue) and defective (orange) viral genomes from three chronically-infected individuals who were sampled shortly post-therapy-initiation. **d** Spectrum of defective HIV-1 DNA sequences at the earliest sampling time points available from each of the four FRESH cohort participants

varieties of defective HIV-1 DNA genomes included: Two cases of premature stop codons attributable to substitution mutations at amino acid positions Env 832 and Gag 322, respectively (Pt 4), and 72 cases of large internal deletions (29 from Pt 3 and 43 from Pt 4; Supplementary Table 3, rows 8 and 12). In summary, viral genome defects at stage V were different in composition compared to stage II samples and included predominantly large deletions (83% (29/35) and 90% (43/48) in Pt 3 and 4), followed by APOBEC-3G/3F-associated hypermutations (6% (2/35) and 6% (3/48)) and premature stop codons from substitution mutations (0% (0/35) and 4% (2/48)).

**Longitudinal evolution of HIV-1 DNA after acute infection.** Overall, this study collected a total of 292 HIV-1 DNA sequences in all four patients sequentially sampled over a 1-year follow-up period (Fig. 4a). Absolute proportions of both genome-intact and genome-defective viruses per million PBMCs sampled showed a general trend of decreasing over the 1-year follow-up duration (Fig. 4b, c). Relative proportions of genome-intact HIV-1 DNA sequences also showed a general trend of decreasing over time, while the proportions of other genome-defective categories varied longitudinally (Fig. 4d). It should be noted that these observations were limited by the low number of HIV-1 genomes that were recovered at multiple time points from the blood samples available for testing (Supplementary Table 3). Among all 87 genome-intact proviruses detected in the four patients during the first year of infection, 100% (Pt 1, 2, 3) and 87% (Pt 4) of the intra-host genetic variations were attributed to single base substitution mutation events; 13% of the intact genomes in Pt 4 contained non-lethal insertion/deletion mutations (Fig. 5a). In both untreated patients, sequence diversity increased over the follow-up duration from a pairwise median of 8 bp differences between intact genomes at baseline to 18 bp differences in Pt 2, and from 4 to 39 bp differences in Pt 4. In contrast, diversity among intact proviral sequences over the follow-up duration decreased and/or remained relatively constant in ART-treated patients from a median of 3 to 6 bp differences in Pt 1, and from 3 bp to undetectable in Pt 3 (Supplementary Table 4). In comparison, in a chronically-infected patient (Pt 6, Supplementary Table 2), pairwise distance between intact viral genomes ranged from 172 to 235 bp; this analysis was not performed for Pt 5 nor Pt 7 because only one intact genome was recovered from each of the two patients. Interestingly, in Pt 1, whose regimen contained raltegravir from the beginning of the antiretroviral treatment initiation, a genome-intact provirus harboring a single-base substitution at 1-month post-detection translated into Y143H, an integrase inhibitor-associated resistance mutation[27] (Fig. 5a). Furthermore, no cases of complete sequence identity between distinct intact or defective viral genomes were detected over the entire 1-year study duration in all sequences derived from Pt 1 (19 sequences from all time points), Pt 2 (27), nor Pt 3 (53). In contrast, three clusters of completely identical HIV-1 DNA sequences were detected in Pt 4 at the time of diagnosis at stage V and at subsequent time points (two clusters consisting of intact genomes marked in Fig. 5a, b as Cluster 1 and 2, and one cluster containing two identical sequences with large deletions, termed Cluster 3). No viral lineage(s) became dominant over time (Fig. 5b).

**Cytotoxic T cell epitope diversity.** Each patient in this study had one protective HLA class I allele known to be statistically associated with protection against disease progression[25,26]. To investigate the influence of HLA class I-dependent immune pressure on HIV-1 DNA sequences, we first defined regions in the viral genomes that contained epitopes restricted by autologous protective HLA class I alleles in each patient, using the Los Alamos HIV Molecular Immunology Database[28] as a reference. Then, we subjected each intact viral genome to a search for sequence variations consistent with CTL-driven escape mutations in the pre-defined epitopes. Specifically, CTL epitopes associated with HLA-B 44:03, 81:01, 57:02, 58:01 in Pt 1–Pt 4, respectively (Table 1), were examined longitudinally for evidence of mutations away from viruses detected at the earliest sampling time points. In total, we evaluated 67 distinct epitopes, including 15 epitopes in Pt 1, 12 in Pt 2, 9 in Pt 3, and 31 in Pt 4 spanning all HIV proteins from Gag to Nef. Over time, we did not observe evidence for viral sequence diversification in CTL epitopes in Pt 1, 2, and 3. In Pt 1

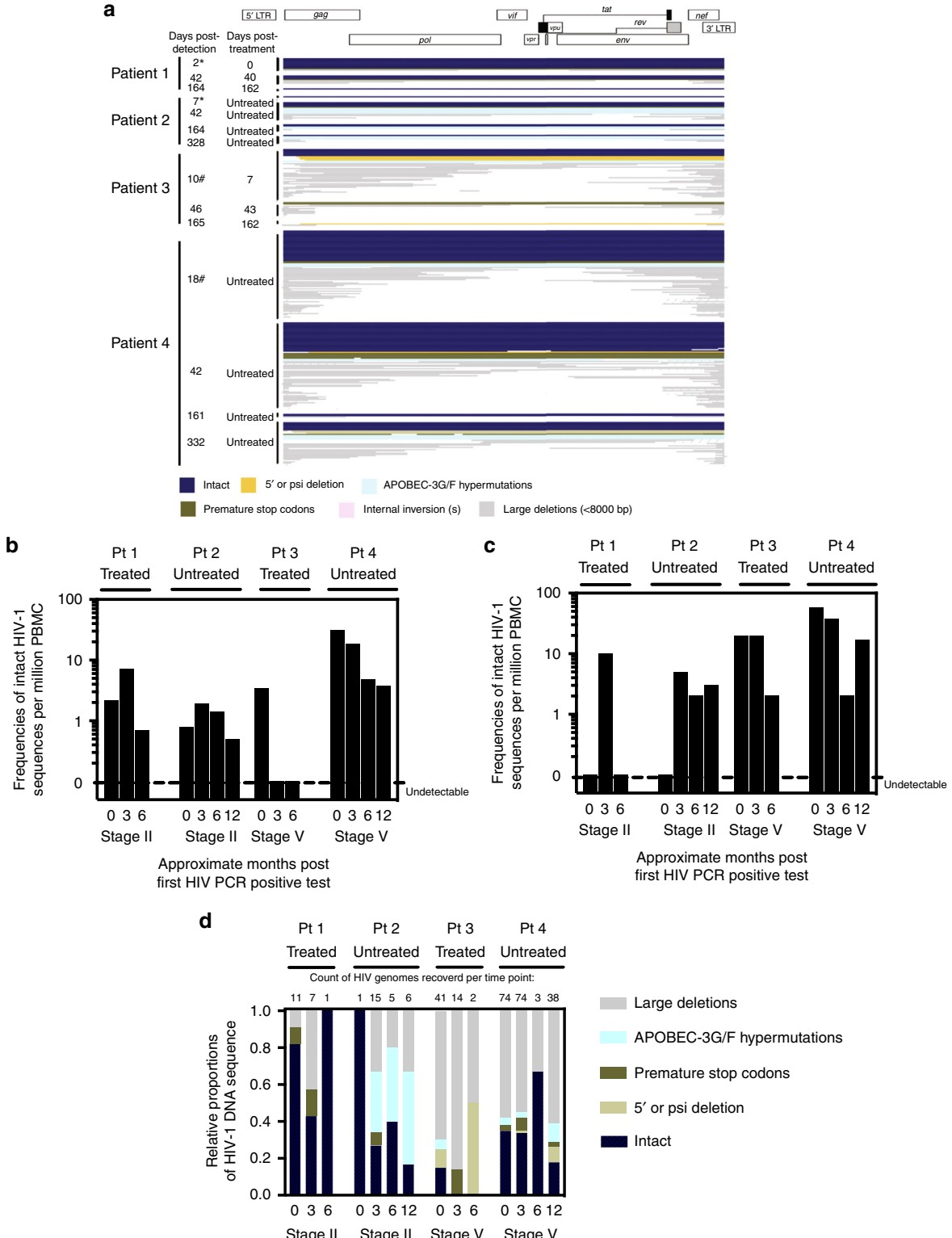

**Fig. 4** Longitudinal changes in HIV-1 DNA genotypic compositions. **a** Diagram reflecting a cross-sectional view of all 292 HIV-1 genomes presented in this study from all four patients over all sampling time points (each horizontal line represents one viral genome). Navy blue lines represent intact HIV-1 genomes; other colors represent defective HIV genomes. Asterisk denotes sampling time points at stage II. Hash denotes sampling time points at stage V. **b** Absolute frequencies of intact HIV-1 DNA sequences per million PBMCs during the 1st year of infection. **c** Absolute frequencies of defective HIV-1 DNA sequences per million PBMCs during the 1st year of infection. **d** Relative proportions of HIV-1 DNA sequences during the 1st year of infection

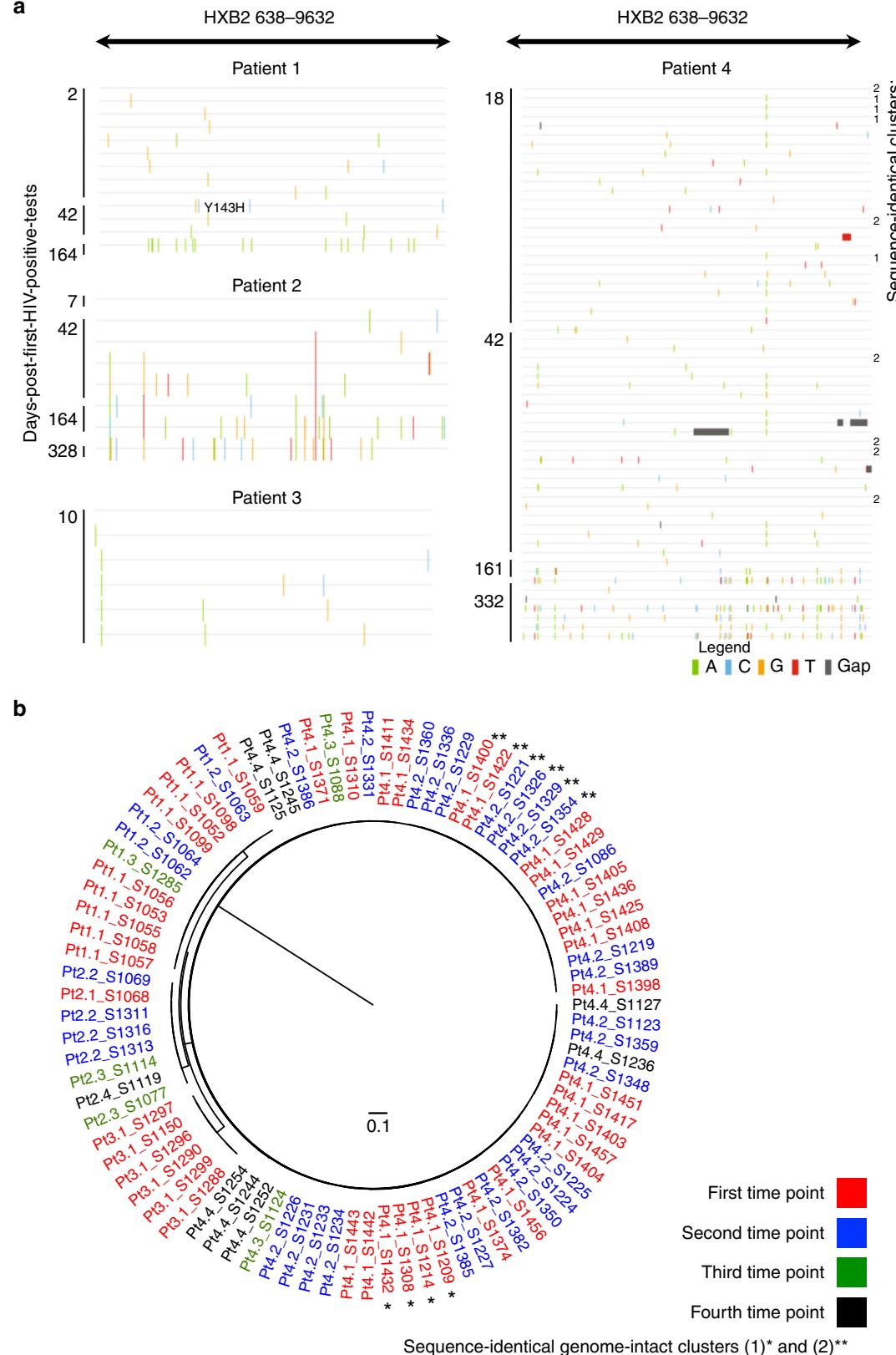

and Pt 2, 194/195 (99%) and 94/96 (98%) epitope sequences remained identical over the 1-year follow-up time, respectively. Also, in Pt 1 and Pt 2, CTL epitope sequences from defective viral genomes shared close to 100% genotypic identity with those derived from intact viral genomes and did not show obvious signs of CTL-driven mutational escape over time. In Pt 3, intact genomes were only detected at the earliest time point and no sequence diversification in CTL epitopes was observed over time.

**Fig. 5** Genetic variations among intact HIV-1 DNA genomes detected in this study. **a** Los Alamos HIV Sequence Database highlighter plots of all intact viral genomes derived in this study across all time points. Each horizontal line represents an intact HIV-1 DNA genome detected in this study spanning HXB2 638-9632. For each patient, a random sequence from the earliest time point was selected to serve as the comparator against the rest of the intra-patient viral genomes (top-most line in each patient, unmarked, master sequence). Vertical strokes represent bases that were different from the master sequence (green A, blue C, orange G, red T, gray gap/deletion). Two genome-intact clonal clusters were detected in Pt 4 and were labeled as 1 and 2 on the right side of the panel. **b** FastTree2 single precision approximately-maximum-likelihood phylogenetic tree of HIV-1 DNA intact genomes. This method was chosen to resolve full-viral-genome sequences with extreme homology; branch lengths were likely inflated. Sequence-identical viral genomes were marked with (*) and (**)

In contrast, three interesting sequence shifts occurred in Pt 4: These three shifts were found exclusively in Nef in regions containing well-defined HLA-B*58:01-restricted epitopes listed in the Los Alamos HIV Molecular Immunology Database[28], including KF9/AL9 (KAAFDLSFF, Nef position 82–90 and overlapping AAVDLSHFL, Nef Position 83–91), YY9 (YTPGPGVRY, Nef position 127–135), and YF9 (YPLTFGWCF, Nef position 135–143). For each of the three epitopes, one new mutated sequence was detected, and all increased in prevalence from 0% (0/26 intact genomes containing escape mutations) at baseline to 71% (5/7), 43% (3/7), and 43% (3/7), respectively, 1 year after infection (Supplementary Table 5). In Pt 4, no other sequence shifts were observed in the other 28/31 (90%) epitopes evaluated.

## Discussion

In this study, we provide a comprehensive evaluation of early HIV-1 DNA genome landscapes during the 1st year of infection. In the context of the four patients examined and the number of cells sampled, we observed that (i) most HIV-1 DNA genomes detectable at stage II were genome-intact, (ii) single-base substitution mutations, likely attributable to errors due to reverse transcriptase activities, were responsible for the majority of early HIV-1 DNA genotypic diversity, (iii) host APOBEC-3G/3F activity, as measured by the presence of hypermutated viral genomes, was not observed in the earliest phase of infection but increased in occurrence over time, (iv) viral genome truncation, the most commonly observed type of defect in chronically-infected patients receiving antiretroviral therapy[7,21,29,30], was rare in the earliest phase of infection but also increased over time, (v) identical intact proviral sequences, suggestive of clonal proliferation of HIV-1-infected CD4 T cells, were uncommon in the earliest stages of infection, and finally, (vi) signs of CTL-mediated immune selection pressure were unexpectedly weak over the first year of infection despite the presence of protective HLA-alleles.

Our study is foremost limited by our sample size and sampling depth: we were only able to sample 0.4–4.1 million PBMCs per sampling time point due to the extreme scarcity of blood samples available from our study patients. Furthermore, each of the four patients examined had unique and protective HLA-alleles, differed in treatment status (treated versus untreated), and also varied in terms of clinical profiles including viral loads and CD4 counts. As such, our observations must be regarded as individual patient profiles and should not be over-generalized. Nonetheless, due to its longitudinal nature, our study may serve as a basis for further investigations of viral DNA evolutionary dynamics during and after acute HIV-1 infection.

Despite our limitations in sample size and sampling depth, this study is, to our knowledge, the first to longitudinally quantify population evolutionary dynamics of HIV-1 DNA species in hyperacute HIV-1 infection, specifically in the context of the HIV-1 clade C epidemic in sub-Saharan Africa. In this context, it is important to emphasize that in this study we describe snapshots of HIV-1 DNA genotypic compositions at the earliest stages of infection; some of these infected cells, but not all, could become future latent reservoirs, defined as infected cells with transcriptionally silenced HIV-1 genomes in patients undergoing

suppressive treatment. In addition, our near-full-length viral genome sequencing method involved PCR amplification spanning HXB2 coordinates 638 (Lys tRNA primer binding site) to 9632 (3′LTR R repeat), and therefore could theoretically amplify both episomal 2-LTR HIV-1 DNA and integrated proviral genomes. It has been shown that 2-LTR circles accumulate early on in acute infections and decline after prolonged antiretroviral treatment[12]. Therefore, we cannot exclude that proportions of the HIV-1 DNA copies reported here represented extrachromosomal rather than proviral HIV DNA.

One of the particularly intriguing findings from this study is the high proportion of genome-intact viruses in the two stage II samples. Such prominence of genome-intact viral genomes is likely explained by active viral replication and suggests that viral genome truncation is a relatively rare event among newly infected CD4+ cells during acute viremia but increases proportion-wise relative to intact viral genomes as the infection progresses. For comparison, we sequenced viral genomes from three chronically-infected patients shortly after therapy initiation, which demonstrated profoundly reduced proportions of genome-intact viruses relative to individuals with acute HIV-1 infection. These data together imply that sustained viremia for prolonged periods of time prior to treatment initiation may have already led to the accumulation of a large number of defective proviruses. This observation is consistent with previous reports by Bruner et al. and Pinzone et al. demonstrating that defective viruses dominate the viral reservoir landscape in chronically-infected patients[7,31]. Further studies should aim to sample patients at much tighter sampling intervals and with a much larger sample size in order to elucidate the decay dynamics of genome-intact viral DNA.

In this study, we also detected no evidence of sequence-identical viruses in three of the four patients. Previous studies have used complete viral genome sequence homology to infer clonal expansion of infected cells in chronically-infected patients[29,30]. However, due to the high level of viral sequence homology observed in this cohort, integration site analysis (using for example a method described in ref. [32]) should be used to examine the presence or absence of clonal expansion during the first year of infection. Nevertheless, the absence of sequence-identical viral sequences in three out of four of our patients may suggest that clonal proliferation of HIV-1-infected CD4 T cells is not a frequent phenomenon during the earliest stages of HIV-1 infection. Of note, this low level of sequence-identical viruses in the FRESH cohort represented a sharp contrast to our previously published results from chronically-treated virologically-suppressed participants, in which 12/26 (46%) PBMC-derived intact HIV-1 DNA genomes fell into sequence-identical, clonal clusters[29].

Another interesting observation in this study is the lack of potent CTL-driven evolution in patients with protective HLA class I alleles, including two patients who did not receive antiretroviral therapy and were presumably exposed to stronger CTL-dependent immune activity. This was demonstrated by the lack of escape mutants in Pt 2 and by the incomplete dominance of mutants in Pt 4. Furthermore, we only observed mutations in HLA-B*58:01-restricted Nef epitopes among all HIV proteins examined, consistent with prior studies demonstrating that Nef is

a predominant target for HIV-1-specific CTL during early stages of HIV-1 infection[33]. The lack of CTL-driven selective pressure outside of Nef during the first year of infection raises important questions about the roles of CTLs for shaping the proviral reservoirs during the course of infection.

In summary, we have provided an in-depth longitudinal evaluation of HIV-1 DNA genome compositions during the first year of infection and have shed light on factors that shaped early HIV-1 DNA genotypic landscapes in four patients. The recent finding that treatment initiation as early as stage I, followed by a median 2.8 years of suppressive treatment, would not eliminate persisting viral reservoirs[8] stresses that an extremely small number of infected cells with genome-intact HIV-1 can result in virologic rebound in the absence of treatment. As such, understanding the properties of this small and obscure reservoir as well as host factors associated with its survival and persistence[34] should be one of the most prioritized goals of HIV-1 reservoir and cure research. In this context, it is important to stress that our study focuses on HIV-1 DNA genotypes sampled during the earliest year of infection; viral and host characteristics of infected cells that survive and persist long-term under suppressive therapy remain undefined, and should be further characterized by longitudinal sampling that extend beyond the time frame of the current study.

## Methods

**Ethics statement**. This study was approved by the Biomedical Research Ethics Committee of the University of KwaZulu-Natal and the Institutional Review Board of Massachusetts General Hospital. All participants provided written informed consent.

**Cohort description**. The FRESH cohort is an observational, prospective cohort launched in 2012 in Umlazi, Durban, South Africa where high-risk HIV-negative women aged 18–23 were recruited to received twice-weekly HIV-1 RNA PCR testing by finger prick during 96 weeks of surveillance, later revised to 48 weeks. The study design incorporated a socioeconomic empowerment intervention to address challenges that affect young women in this setting that may contribute to increased HIV acquisition risk[23,24,35]. Following detection of acute HIV-1 infection, PBMCs were collected at weekly intervals for a month, then every 2 weeks for 2 months and monthly thereafter for 1-year post infection. As of August 2018, 946 women were enrolled; 72 were identified with acute HIV-1 infection. "Acute HIV infection" was defined as a new plasma HIV-1 RNA detection, that was positive on repeat testing with an evolving HIV-1 Western blot pattern (prior to development of a p31 band). Acute infection staging according to the classification of Fiebig et al.[22] was based on plasma HIV-1 RNA, plasma p24 antigen (p24 Ag), fourth generation HIV enzyme immunoassay (EIA) and Western blot (WB) and defined as follows: stage I (RNA+, p24 Ag−, EIA−, WB−); stage II (RNA+, p24 Ag+, EIA + or −, WB-); stage III (RNA+, p24 Ag+ or −, EIA+, WB- or indeterminate); stage IV (RNA+, p24 Ag+ or −, EIA+, WB indeterminate or +); stage V (RNA+, p24 Ag+ or −, EIA+, WB+ without p31 band)[22,35,36] (Supplementary Table 1). Peak viremia was defined as the highest viral load detected during the initial viral load increase/decline stage. No participants were recruited for this specific study; banked, cryopreserved PBMC samples were used.

**Single-template viral genome amplification**. Total PBMC collected from each patient at each time point were independently subjected to DNA extraction using DNeasy Blood & Tissue Kits (QIAGEN). Total HIV-1 and host cell concentrations in the DNA extracts were estimated using BIO-RAD ddPCR, using primers and probes covering HIV-1 5′ LTR-gag HXB2 coordinates 684–810[37] (forward primer 5′-TCTCGACGCAGGACTCG-3′, reverse primer 5′-TACTGA CGCTCTCGCACC-3′ probe/56-FAM/CTCTCTCCT/ZEN/TCTAGCCTC/ 31ABkFQ/, and human RPP30 gene[38] forward primer 5′-GATTTGGACCTGC GAGCG-3′, reverse primer 5′-GCGGCTGTCTCCACAAGT-3′, probe/56-FAM/ CTGACCTGA/ZEN/AGGCTCT/31ABkFQ/). ddPCR was performed using the following thermocycler program: 95 °C for 10 min, 45 cycles of 94 °C for 30 s and 60 °C for 1 min, 72 °C for 1 min. The droplets were subsequently read by the BIO-RAD QX100 droplet reader and data were analyzed using QuantaSoft software (BIO-RAD). Extracted DNA was diluted according to ddPCR results and Poisson distribution statistics, so that the statistical probability of one HIV-1 genome template per PCR reaction was 85.7%. This was followed by HIV-1 near-full-genome amplification using a single-amplicon nested PCR approach[29,39] (first-round nested-PCR: forward primer 5′-AAATCTCTAGCAGTGGCGCCCGAA CAG-3′, reverse primer 5′-TGAGGGATCTCTAGTTACCAGAGTC-3′; second-round nested-PCR: forward primer 5′-GCGCCCGAACAGGGACYTGAAA

RCGAAAG-3′, reverse primer 5′-GCACTCAAGGCAAGCTTTATTGAGGCT TA-3′). One unit of Platinum Taq (Invitrogen) per 20 µL reaction was incubated with 1× reaction buffer, 2 mM MgSO$_4$, 0.2 mM dNTP, 0.4 µM each of forward and reverse primers, and subjected to the following PCR program: 2 min at 92 °C, 10 cycles [10 s at 92 °C, 30 s at 60 °C, 10 min at 68 °C], 20 cycles [10 s at 92 °C, 30 s at 55 °C, 10 min at 68 °C], 10 min at 68 °C, 4 °C infinite hold. Figure 2a illustrates the sample processing workflow.

**MiSeq (Illumina) deep sequencing and viral bioinformatics**. All PCR amplicons detectable by gel electrophoresis were subjected to Illumina library preparation via shearing-based adaptor-ligation-based technology and MiSeq sequencing, followed by de novo assembly of the resulting small reads in collaboration with the Massachusetts General Hospital CCIB DNA Core. In this study, consensus FASTA sequences derived from assembled contigs were screened for defects in the viral genomes via an in-house sequence analysis pipeline termed HIVSeqinR (version 2.6). Figure 2b outlines the bioinformatics workflow. The first key step in HIV-SeqinR was the mapping of each viral genome against the reference genome HXB2 with local megablastn from the NCBI blast+ suite[40,41] to remove any non-HIV sequences derived from non-specific amplifications at the PCR step. Since mega-blast only extends an alignment when there is an exact match of 28 base pairs (bp), it would therefore detect junctions of any internal genome breakages; in contrast, high-diversity regions such as the variable regions in *env* may not map. To accommodate for such natural HIV genotypic diversity, we defined any sequence to be HIV if ≥80% of the total contig length mapped to HXB2. Subsequently, viral genome defects were identified in the order of (i) large internal deletion(s) defined as sequence length <8000 bp (LargeDeletion), (ii) internal inversions or genome-scrambling without inversions (InternalInversion or Scramble), (iii) APOBEC-3G/ 3F-associated hypermutations (Hypermut) using a within-pipeline adaptation of the Los Alamos Hypermute 2.0 web tool[42], (iv) presence of premature stop codons and/or insertions or deletions in any of HIV's essential genes *gag*, *pol* or *env* that would result in amino acid sequences <95% or >120% of expected lengths relative to HXB2 (PrematureStop), and (v) indels at the 5′ packaging signal region that accumulatively exceeded 15 bp relative to HXB2 (5DEFECT). Any HIV-1 proviral genomes that do not have any of these pre-defined defects are classified as "Intact." All other sequences that did not satisfy the above defectiveness classes were marked "Check" to alert users to manually examine the genome. Finally, a key feature of HIVSeqinR is our method used to translate HIV-1 genes. For each gene, individual gene regions are defined by recording flanking start and stop codon coordinates guided by gene-specific nucleotide alignment; resulting sequences are subsequently degapped before translation. This method serves to avoid mis-translation that arises from common cases such as GG-CCCAA-TT which would be at risk for false translation to X-Pro-X-X (despite usage of codon alignment) instead of the correct translation to Gly-Pro-IIe. All multiple sequence alignments in this study were performed using MUSCLE[43] or MAFFT[44]. Phylogenetic distances between sequences were examined using FastTree 2.1.10 approximately-maximum-likelihood single precision phylogenetic trees[45] and/or Hamming distances calculated using the R-package Biostrings[46]. Primer binding sites were excluded from analyses. All statistical tests and other genetic distance analyses in this study were performed with R[47] and Prism7 (GraphPad Software).

**Reporting summary**. Further information on research design is available in the Nature Research Reporting Summary linked to this article.

## Data availability

All raw HIV sequence data that support the findings of this study have been deposited in GenBank (accession numbers MK643536–MK643827 [https://www.ncbi.nlm.nih.gov/WebSub/?form=dwnld&sid=2204500&tool=genbank]). Data underlying all figures are reported in Supplementary Tables 2, 3, and 4. The authors declare that all data supporting the findings of this study are available within the paper and its Supplementary Information files.

## Code availability

The R-script of HIVSeqinR version 2.6 and all its documentations are freely available at Github (https://github.com/guineverelee/HIVSeqinR). The output of HIVSeqinR consists of six key files exported into the folder Results_Final: (i) a comma delimited (CSV) file that contains translations of each gene from each sample, (ii) a CSV file that contains the final verdict for each contig (genome-intact or defective; type of defects), (iii) a FASTA file containing all pairwise alignments of each viral gene against HXB2 in the nucleotide space, (iv) a similar FASTA file in amino acid space, (v) a PDF representation of any contigs that are ≥8000 base pairs (bp) and did not have any hypermutations nor internal inversions, (vi) a multiple sequence alignment of all HIV contigs that are ≥8000 bp for users to visually examine any unanticipated abnormalities. HIVSeqinR has been validated to process subtype B and C HIV-1 DNA sequences.

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

## Acknowledgements

M.L. is supported by NIH grants AI114235, AI117841, AI120008, AI130005, AI122377, AI124776, and AI135940. X.G.Y. is supported by NIH grants DA047034, HL134539, and AI116228. The study cohort and sample collection were supported in part by grants from the Bill and Melinda Gates Foundation (OPP1066973 and OPP1146433), Gilead Sciences, Inc. (Grant ID #00406), the International AIDS Vaccine Initiative (IAVI) (UKZNRSA1001), the NIAID (R37AI067073), the Witten Family Foundation, the Dan and Marjorie Sullivan Foundation, the Mark and Lisa Schwartz Foundation, Ursula Brunner, the AIDS Healthcare Foundation, and the Harvard University Center for AIDS Research (CFAR, P30 AI060354, which is supported by the following institutes and centers co-funded by and participating with the US National Institutes of Health: NIAID, NCI, NICHD, NHLBI, NIDA, NIMH, NIA, FIC, and OAR.). Raltegravir used for immediate treatment was donated by Merck & Co., Inc. This work was also partially supported through the Sub-Saharan African Network for TB/HIV Research Excellence (SANTHE), a DELTAS Africa Initiative [Grant # DEL-15-006]. The DELTAS Africa Initiative is an independent funding scheme of the African Academy of Sciences (AAS)'s Alliance for Accelerating Excellence in Science in Africa (AESA) and supported by the New Partnership for Africa's Development Planning and Coordinating Agency (NEPAD Agency) with funding from the Wellcome Trust [Grant # 107752/Z/15/Z] and the United Kingdom (UK) government. The views expressed in this publication are those of the author(s) and not necessarily those of AAS, NEPAD Agency, Wellcome Trust, or the UK government. The authors thank all patients participating in the FRESH cohort whom have made this study possible. The authors thank the Massachusetts General Hospital Center for Computational & Integrative Biology DNA Core, specifically Dr. Nicole Stange-Thomann, Dr. Amy Avery, Ms. Kristina Belanger, and Mr. Huajun Wang, for providing them with the Illumina MiSeq deep sequencing service used in this manuscript. The authors also thank Dr. Art Poon (Western University, Ontario, Canada) for his advice on viral phylogenetic analyses.

## Author contributions

The work presented here was carried out in collaboration between all authors. The study was conceptualized and designed by G.Q.L., K.R., K.G., T.N., X.G.Y. and M.L. PBMC samples and clinical/demographic data were collected by K.L.D., B.D.W., K.R., K.G., and T.N. HIV-1 genotyping laboratory work was done by G.Q.L., K.R., K.G., K.B.E. and J.M.C. Results were analyzed by G.Q.L., K.R., K.G., K.B.E., J.M.C., X.G.Y., T.N. and M.L.

G.Q.L. wrote HIVSeqinR. G.Q.L. and M.L. wrote the manuscript; all authors contributed to and approved the manuscript. M.L. supervised the study.

## Additional information

**Competing interests:** M.L. has received speaking and consulting honoraria from Merck & Co., Inc. and Gilead Sciences Inc. T.N. receives research funding support from Gilead Sciences Inc., a pharmaceutical company with interests in HIV cure research. The other authors declare no competing interests.

