## [Peer Review File · Nature Communications]

Reviewers' comments:

Reviewer #1 (Remarks to the Author):

The authors have carried out full length single genome sequencing of HIV proviruses in PPMC of 4 patients identified in very early stages of HIV infection. They find a higher fraction of intact sequences than has been observed in patients treated at later stages of infection and an absence of CTL escape mutations. The study is said to provide a description of the HIV reservoir at early time points after infection.

A major problem with the study is that many of the samples were taken from time points at which the patients were viremic or recently viremic. There is ample evidence that viremia and infected cell frequencies decline rapidly in the months after ART initiation. Eventually, at 6 months to 1 year after ART initiation, a stable level of infected cells is reached. These cells constitute the long term reservoir for HIV. For this reason, most studies of the reservoir only include samples from patients who have been on ART for a minimum of 6 months. The authors appears to be aware of this issue, but continue to use the term reservoir in an incorrect way. If the authors wish to study the reservoir, then the analysis should be restricted to samples taken at least 6 months after initiation of treatment. It appears that if earlier sequences are excluded, the resulting data set may not differ much from previous studies. As a control, it would be helpful to see similar sequencing on patients who start ART during chronic infection and are sampled at short times after initiation of ART. It is likely that a higher fraction of intact proviruses would also be observed in that setting.

Reviewer #2 (Remarks to the Author):

Lee and colleagues, in their manuscript entitled "HIV-1 Reservoir Landscapes in Acute Subtype C Infections", are examining the integrated HIV sequences in four individual patients from the FRESH South African Cohort of high-risk women, to gain better insight into early sources of integrated viral genomes. The main finding of the study is that most of the genomes examined were intact and mutation/indel-free, and that even after following the patients for some time, only minimal mutations arose, including those that may be expected to result in abrogation of CTL recognition. Furthermore, the authors conclude that the treated vs. untreated patients had similar levels of viral reduction after 1 year.

Strengths of the study: The team and the cohort are extremely well positioned to carry out this study, which I agree with the authors, addresses an underexamined niche in the field, that being the nature and evolution of the very early viral genome integrands. The authors have also developed a highly sensitive method for assembling individual viral genome sequences through PMBC amplification and NGS, and this is a marvelous methodological platform, and is no easy technical feat. The lack of mutations, and minimal contribution of APOBECs, and finding mostly of point mutations are novel and significant.

Weaknesses of the study: Unfortunately, as presented the manuscript has a number of critical issues.

- 1) The critical shortcoming is extremely limited / borderline insignificant Sample size: Only 4 patients were used in the whole study, and within the 4 patients, there is diversity in HLA (of course to be expected: in the beginning it stated that all 4 have protective HLA alleles, but later it is stated that one also has a fast progression-related HLA), but also diversity in stage (II or V), and as regards treatment (2 treated and 2 untreated, and even differences in the drug regimen between the 2 treated individuals), times of sampling, levels of viremia etc. Thus, looking at only 4 patients with almost every variable that can vary indeed significantly varying, I am extremely hesitant to support the absolute conclusions made by the authors on the basis of this sample size, especially as reflected by the small sample size, a lot of time, the percentages given for certain phenomena are "0%" In my mind, such conclusions (many of which go against the field and could have very important implications) require larger sample sizes, and statistical analyses. In addition, it is unclear what the rationale was for picking these particular 4 patients. I suspect it may have to do with carrying protective HLA alleles, such that the authors hoped to find CTL-related mutations, but this is not described.
- 2) Clonal expansion studies assume that different viral genome sequences mean different clones, but the major issue with this is that the authors have already shown there is little sequence diversification in their study cohort; based on this, the logic that the same sequence implies a clone is flawed.
- 3) The section on receptor tropism stands out as an odd section in this manuscript and it does not follow the rest of the story and is perhaps suited for a different manuscript.
- 4) The authors posit that ddPCR overestimates reservoir sizes, but it is not clear why one method is taken to provide the "true" size and the other as the inaccurate one.

5) As written, many of the sections are descriptive. There are often exceptions...for instance when 1 patient differed from the other 3 in some aspect and this is thoroughly mentioned, but the significance/conclusions of such individual variations are not clear: reads like a case report

6) I found numerous writing issues, which in general gave the manuscript a rather unpolished feel:

-e.g. putting %s in the title of a section, numerous grammar issues (plural vs. singular), and could use another round of proof-reading for quality of the writing

-Some of the figures have an unpolished feel as well, for e.g. Figure 2A and 2C have numerous gross misalignments of graph bars, etc.

-The authors use the term "hypermutation" as regards APOBECs but they are referring to very low mutations, so this term is incorrect

-The authors use the term "defect" or "defective" for almost everything that is not the full wildtype unmutated sequence, but I am not sure this is appropriate as mutated or even truncated genomes can still lead to infectious virus some of the time, or at least produce proteins that can be recognized by the immune system.

-Throughout the manuscript, the authors make many absolute statements (such as absolute percentages or time-lines), which are not in fact established as absolutes (either in their own data due to above-mentioned limitations in sample size, or in the published literature – e.g. the statement that the reservoir has a half-life of exactly 44 months)

-Some statements are not backed up by the literature: for instance, in the discussion the authors present the notion that perhaps CTL epitope mutants were not detected because there is something wrong with the peptide/MHC presentation pathway. This is entirely conjecture without any evidence, and there is no evidence that such can happen. It is more plausible (as the authors also recognize) that there is an issue with transcription.

Reviewer #3 (Remarks to the Author):

The authors apply assays that they have previously described for chronically infected subjects with subtype B infection. These are new and interesting data from the very interesting FRESH cohort, although the data are limited to only 4 subjects (and only 1 sequence in patient 2).

The data are of interest, because this sort of study has not been done before; however, the ability to say too much more than the descriptive presentation of the data is limited by the fact that we only have 2 pairs of subjects and one subject is an elite controller (unlike 99% of most subjects with HIV infection). For example, “Pharmacological vs natural HIV-1 control resulted in similar level of reservoir size reductions over one-year follow up” make assertions that do not seem robust with these 4 subjects.

Regarding “clonal expansion” discussed on the bottom of page 12, with a monoclonal viral population during acute infection, as the authors know, infection of 2 cells by the progeny of a monophyletic infection or proliferation of a single infected cell cannot be distinguished without integration site analysis.

The tropism study is a bit concerning, since 1) the phenotypic correlations to validate tropism algorithms using sequence are much less robust for non-B than for B subtypes, and 2) even with subtype B most publications have been expected to use a 5% FPR.

In the last sentence of the Abstract, what are the mechanistic insights? The observations are descriptive.

Finally, the text really should be revised by someone who can do justice to the data by making the text easier to read with clearer and more correct prose. One example is the sentence in lines 185-187, but this sort of sentence detracts from the text throughout. Throughout, punctuation, abbreviations, grammar and word usage are sloppy. Also, usage of nouns as adjectives impairs the text, but I have elected not to detail these.

The senior authors should put in more effort in revising the text.

Point-by-point Response to the Reviewers' Comments

Reviewer #1 (Remarks to the Author):

The authors have carried out full length single genome sequencing of HIV proviruses in PBMC of 4 patients identified in very early stages of HIV infection. They find a higher fraction of intact sequences than has been observed in patients treated at later stages of infection and an absence of CTL escape mutations. The study is said to provide a description of the HIV reservoir at early time points after infection.

A major problem with the study is that many of the samples were taken from time points at which the patients were viremic or recently viremic. There is ample evidence that viremia and infected cell frequencies decline rapidly in the months after ART initiation. Eventually, at 6 months to 1 year after ART initiation, a stable level of infected cells is reached. These cells constitute the long term reservoir for HIV. For this reason, most studies of the reservoir only include samples from patients who have been on ART for a minimum of 6 months. The authors appears to be aware of this issue, but continue to use the term reservoir in an incorrect way. If the authors wish to study the reservoir, then the analysis should be restricted to samples taken at least 6 months after initiation of treatment.

Response: We fully agree with Reviewer #1 that our usage of the term “reservoir” was not appropriate in the context of this study, and fully acknowledge that only a small portion of these viral DNA genomes will persist after ART-initiation and virologic suppression to become stable “reservoirs.” In response, we have extensively revised the manuscript to reflect what we describe in this study were HIV-1 DNA sequences sampled during the first year of infection.

It appears that if earlier sequences are excluded, the resulting data set may not differ much from previous studies. As a control, it would be helpful to see similar sequencing on patients who start ART during chronic infection and are sampled at short times after initiation of ART. It is likely that a higher fraction of intact proviruses would also be observed in that setting.

Response: We thank Reviewer #1 for this very helpful suggestion, and fully agree that samples obtained shortly after initiation of ART in patients who started antiretroviral therapy during chronic infection would serve as appropriate controls for our observations. In this revised manuscript, we have now added sequencing results of three chronically-infected patients who were sampled shortly after therapy initiation. In sum, we have included 286 additional HIV-1 DNA sequences as comparators against the FRESH cohort for this revision.

In the FRESH cohort, the percentage intact genomes in PBMC obtained during Fiebig stage II were 82% and 100% (Pt 1 and Pt 2) versus 15% and 35% at Fiebig stage V (Pt 3 and Pt 4). Longitudinally, proportions of intact genomes at 0, 1, 6, 12-months post-detection from the four FRESH cohort participants described in this study are listed

below. Numbers within parenthesis indicate the total numbers of intact and defective HIV-1 DNA genomes recovered from each sampling time point (sampling depth). 12-month samples were not available from two of the four patients.

Pt 1: Fiebig stage II at detection, ARV+	82% (11), 43% (7), 100% (1)
Pt 2: Fiebig stage II at detection, ARV-	100% (1), 27% (15), 40% (5), 17% (6)
Pt 3: Fiebig stage V at detection, ARV+	15% (4), 0% (14), 0% (2)
Pt 4: Fiebig stage V at detection, ARV-	35% (74), 34% (74), 67% (3), 18% (38)

It is important to note that all four patients have protective HLA-alleles.

In comparison, the proportion of intact viral DNA genomes observed at 18, 70 and 182-days post-ART-initiation in PBMCs of three chronically-infected individuals who had sustained level of viremia for >4, >7, and >12 years pre-therapy were 1% (175), 5% (64), and 2% (47).

It is striking that the proportion of intact viruses in these three chronically-infected individuals shortly after they received treatment (1%, 5%, 2%) were lower than proportions observed in the FRESH cohort participants at Fiebig stage II (82%, 100%), Fiebig stage V (15%, 35%), and one-year post-infection in the two untreated patients (17%, 18%). This could suggest that years of viremia may have led to an accumulation of defective proviruses in the chronically-infected patients. All the additional data have been incorporated into the revised manuscript. See results, discussion, and Supplementary Table 2.

Reviewer #2 (Remarks to the Author):

Lee and colleagues, in their manuscript entitled “HIV-1 Reservoir Landscapes in Acute Subtype C Infections”, are examining the integrated HIV sequences in four individual patients from the FRESH South African Cohort of high-risk women, to gain better insight into early sources of integrated viral genomes. The main finding of the study is that most of the genomes examined were intact and mutation/indel-free, and that even after following the patients for some time, only minimal mutations arose, including those that may be expected to result in abrogation of CTL recognition. Furthermore, the authors conclude that the treated vs. untreated patients had similar levels of viral reduction after 1 year.

Strengths of the study: The team and the cohort are extremely well positioned to carry out this study, which I agree with the authors, addresses an underexamined niche in the field, that being the nature and evolution of the very early viral genome integrands. The authors have also developed a highly sensitive method for assembling individual viral genome sequences through PMBC amplification and NGS, and this is a marvelous methodological platform, and is no easy technical feat. The lack of mutations, and minimal contribution of APOBECs, and finding mostly of point mutations are novel and significant.

Weaknesses of the study: Unfortunately, as presented the manuscript has a number of

critical issues.

1) The critical shortcoming is extremely limited / borderline insignificant Sample size: Only 4 patients were used in the whole study, and within the 4 patients, there is diversity in HLA (of course to be expected: in the beginning it is stated that all 4 have protective HLA alleles, but later it is stated that one also has a fast progression-related HLA), but also diversity in stage (II or V), and as regards treatment (2 treated and 2 untreated, and even differences in the drug regimen between the 2 treated individuals), times of sampling, levels of viremia etc. Thus, looking at only 4 patients with almost every variable that can vary indeed significantly varying, I am extremely hesitant to support the absolute conclusions made by the authors on the basis of this sample size, especially as reflected by the small sample size, a lot of time, the percentages given for certain phenomena are “0%” In my mind, such conclusions (many of which go against the field and could have very important implications) require larger sample sizes, and statistical analyses.

Response: We thank Reviewer #2 for appreciating the technical difficulties of our methods, the rarity of our samples, and the potential significance of our observations. We fully agree with Reviewer #2 that our sample size of four participants with their diverse HLA and clinical profiles were limiting, and that our observations cannot be generalized nor can be phrased as absolute statements. We have extensively revised our manuscript:

Specifically, (i) throughout the manuscript, we have now added limiters to our observations, such as, “In the context of the four patients examined and the number of cells sampled, we observed...” (ii) in the results section, we have revised the manuscript so that all percentages reported are now accompanied with the number of sequences sampled in parentheses to ensure that readers will be aware of the sampling depths behind each frequency calculation. We have also removed any reports of 0% when less than 10 sequences were sampled. (iii) We have now added a paragraph into the discussion section to stress that our conclusions are limited by our sample size and sampling depth, and that a study with larger number of participants will be needed to draw generalizations.

In addition, it is unclear what the rationale was for picking these particular 4 patients. I suspect it may have to do with carrying protective HLA alleles, such that the authors hoped to find CTL-related mutations, but this is not described.

Response: Participants with protective HLA-genotypes were specifically selected to shed light on the dynamics of cytotoxic T cells (CTLs) driven viral evolution over time. We have now clarified our rationale for selecting these four study participants with protective HLA alleles in the results section.

2) Clonal expansion studies assume that different viral genome sequences mean different clones, but the major issue with this is that the authors have already shown

there is little sequence diversification in their study cohort; based on this, the logic that the same sequence implies a clone is flawed.

Response: We acknowledge that 100% sequence homology between sequence pairs alone is insufficient as evidence for clonal expansion due to the limited sequence diversification in this cohort. In this revision, we have removed our claims about clonal expansion, and instead reported the presence of three sequence-identical clusters in viral genomes derived from Pt 4 as mere observations. We have also added a paragraph into the discussion section to point out that integration site analysis is needed to examine clonal expansion of infected cells in these samples.

3) The section on receptor tropism stands out as an odd section in this manuscript and it does not follow the rest of the story and is perhaps suited for a different manuscript.

Response: We agree with both Reviewer #2 and Reviewer #3 that the receptor tropism analysis section stood out as odd, and that there is currently no robust genotypic prediction algorithm to infer non-subtype-B viral tropism using V3 sequence data. In this revision, we have removed the tropism-related paragraphs in both the results and discussion sections.

4) The authors posit that ddPCR overestimates reservoir sizes, but it is not clear why one method is taken to provide the “true” size and the other as the inaccurate one.

Response: Compared to ddPCR which indistinguishably measures both intact and defective viral genomes, viral genome sequencing and bioinformatic inferences of “genome-intact” HIV-1 precludes defective viral genomes and therefore has enabled us to reach a closer estimate of true replication competent viral genome burden. We have now clarified our rationale in the results section; please see edits in the result section: “However, this approach for viral DNA quantification relies on amplification of a short segment of the HIV-1 genome and does not allow to accurately quantify the frequency of intact proviral sequences which may evolve into functionally-relevant components of the long-term HIV-1 reservoir. In fact, the majority of viral sequences identified by ddPCR represent genome-defective HIV-1 DNA products that result from the high error rate of the viral reverse transcriptase and account for a considerable proportion of HIV-1 DNA sequences detectable in individuals undergoing antiretroviral therapy [7,21]. To address this, we performed single-genome, near-full-length next-generation HIV-1 DNA sequencing, followed by a complex biocomputational analysis procedure (Figure 1), to specifically quantify relative proportions of genome-intact HIV-1 DNA sequences and viral DNA species exhibiting genome defects precluding viral replication.”

5) As written, many of the sections are descriptive. There are often exceptions...for instance when 1 patient differed from the other 3 in some aspect and this is thoroughly mentioned, but the significance/conclusions of such individual variations are not clear: reads like a case report

Response: We acknowledge that our sample size of four participants do not allow us to draw generalizations. As mentioned above in response to Reviewer #2's first comment, we have carefully revised the manuscript to remove claims that generalize. We have also added a paragraph in our discussion section to highlight that this study is severely limited by sample size and sampling depth.

We hope the reviewers and the editors understand that it had been extremely rare to obtain samples as early as Fiebig stage II in addition to obtaining longitudinal follow-up samples for one year. Therefore, we believe that this report is unique in providing high-resolution, genome-by-genome insights into the dynamics of the HIV infection processes that has never been reported before.

6) I found numerous writing issues, which in general gave the manuscript a rather unpolished feel:

-e.g. putting %s in the title of a section, numerous grammar issues (plural vs. singular), and could use another round of proof-reading for quality of the writing

Response: The entire manuscript has now been reviewed and revised to improve writing quality. All percentages in the titles of sections have been removed.

-Some of the figures have an unpolished feel as well, for e.g. Figure 2A and 2C have numerous gross misalignments of graph bars, etc.

Response: All figures have been revised and polished.

-The authors use the term "hypermutation" as regards APOBECs but they are referring to very low mutations, so this term is incorrect

Response: We apologize for this confusion. In our manuscript, the term "APOBEC-associated hypermutated genomes" refers to viral genomes that contained excessive number of G-to-A mutations specifically at APOBEC-associated sites, which were deemed statistically significantly associated with APOBEC-3G/F hypermutations by the Los Alamos Webtool "Hypermot 2.0" (p-values less than 0.05). We did not observe this category of genome-defect in samples derived from Fiebig stage II.

In contrast to APOBEC-3G/F-associated hypermutated genomes, another class of mutation we described in the manuscript were "single-base substitution mutations." These mutations were not limited to G-to-A mutations, were not flagged as statistically significant by the Los Alamos Webtool 2.0, and only rarely resulted in genome-defects as defined by having premature stop codons in essential genes. Figure 5A shows that this type of mutation ("single-base substitution mutation") is the main driver behind genetic variations among intact viral genomes during the first year of HIV infections.

We have now clarified our definitions in both the Methods and the Results sections. Specially, in Methods, under "MiSeq (Illumina) Deep Sequencing and Viral

Bioinformatics,” we have now included our definition of APOBEC-3G/3F-associated hypermutated sequences. Our full bioinformatic analyses pipeline to determine viral genome intactness is now included in this revised manuscript. The script is freely available for download at <https://github.com/guineverelee/HIVSeginR>.

-The authors use the term “defect” or “defective” for almost everything that is not the full wildtype unmutated sequence, but I am not sure this is appropriate as mutated or even truncated genomes can still lead to infectious virus some of the time, or at least produce proteins that can be recognized by the immune system.

Response: We again apologize for the confusion. As mentioned above, we have now incorporated into this revised manuscript our entire bioinformatic analyses pipeline and the full definitions of each viral genome defect category. Our definitions adhere to the standard definitions of “defective viral genomes” used in the field (See examples in Ho, Cell, 2013 and Hiener Cell Report 2017). Specifically, “defective genomes” in this manuscript refers to viral DNA genomes that have one or more of (1) truncated genomes that had over ~1000 base pair deletion relative to reference genome HXB2, (2) presence of internal inversions and/or other forms of genome shuffling, (3) presence of APOBEC-associated hypermutations as determined by Los Alamos Webtool Hypermut 2.0, (4) presence of premature stop codons in any one of the essential genes (*gag*, *pol* or *env*), (5) potentially deleterious insertion or deletion mutations in the 5’ psi packaging signal. If a viral DNA genome lacks all of the above defects, it would be deemed “genome-intact.” We would like to highlight that in addition to this computerized sequence analysis, sequences classified as “genome-intact” were manually reviewed for sequences integrity. The sensitivity of the bioinformatic analyses pipeline was cross-validated by sequencing qVOA-derived outgrowth viruses.

-Throughout the manuscript, the authors make many absolute statements (such as absolute percentages or time-lines), which are not in fact established as absolutes (either in their own data due to above-mentioned limitations in sample size, or in the published literature – e.g. the statement that the reservoir has a half-life of exactly 44 months)

Response: We have carefully revised the entire manuscript to remove absolute statements and any inappropriate generalizations. Furthermore, in this revision, we have made sure that absolute percentages or timelines mentioned in this manuscript are always accompanied by limiters.

-Some statements are not backed up by the literature: for instance, in the discussion the authors present the notion that perhaps CTL epitope mutants were not detected because there is something wrong with the peptide/MHC presentation pathway. This is entirely conjecture without any evidence, and there is no evidence that such can happen. It is more plausible (as the authors also recognize) that there is an issue with transcription.

Response: We have now removed these sentences.

Reviewer #3 (Remarks to the Author):

The authors apply assays that they have previously described for chronically infected subjects with subtype B infection. These are new and interesting data from the very interesting FRESH cohort, although the data are limited to only 4 subjects (and only 1 sequence in patient 2).

The data are of interest, because this sort of study has not been done before; however, the ability to say too much more than the descriptive presentation of the data is limited by the fact that we only have 2 pairs of subjects and one subject is an elite controller (unlike 99% of most subjects with HIV infection). For example, “Pharmacological vs natural HIV-1 control resulted in similar level of reservoir size reductions over one-year follow up” make assertions that do not seem robust with these 4 subjects.

Response: We thank you Reviewer #3 for his/her constructive comments. We agree that our sample size is low and that our conclusions cannot be easily generalized. In this revised manuscript, we have carefully removed all attempts to generalize our observations based on these four patients. Specifically, we have removed the section, “Pharmacological vs natural HIV-1 control resulted in similar level of reservoir size reductions over one-year follow up” and have replaced it with a descriptive account of the changes in HIV-1 DNA genotypic compositions overtime in these four patients. See result section, “Longitudinal evolution of HIV-1 DNA sequences after primary HIV-1 infection.”

As noted by Reviewer #3, this sort of study has not been done before; in fact, obtaining samples from days within infection and sustaining a longitudinal sampling of such individuals for full viral genome deep sequencing is one of the key merits of this study. Due to the rare nature of these longitudinal samples and the technical challenges compounded with the high cost of full viral genome sequencing, we believe this study provides insights that are valuable to our understanding of HIV persistence despite the small sample size.

Regarding “clonal expansion” discussed on the bottom of page 12, with a monoclonal viral population during acute infection, as the authors know, infection of 2 cells by the progeny of a monophyletic infection or proliferation of a single infected cell cannot be distinguished without integration site analysis.

Response: We fully agree with Reviewer #3 that integration site analysis is needed to assert this claim. We have now removed the section on clonal expansion in the result section, and have added a paragraph in the discussion to point out the need of integration site analysis if clonal expansion of infected cells were to be examined. Please refer also to our response to Reviewer #2, comment (2).

The tropism study is a bit concerning, since 1) the phenotypic correlations to validate

tropism algorithms using sequence are much less robust for non-B than for B subtypes, and 2) even with subtype B most publications have been expected to use a 5% FPR.

Response: We agree with Reviewer #3 that algorithms for genotypic inferences of subtype C HIV-1 tropism are less robust. Phenotypic characterization via *env* cloning should be done to properly evaluate viral tropisms in the FRESH cohort. In line with Reviewer #2's comment that this section is out of place and did not contribute to the overall story, we have now removed the section. We will pursue this topic in future studies. Please refer to our response to Reviewer #2, comment (3).

In the last sentence of the Abstract, what are the mechanistic insights? The observations are descriptive.

Response: We have now removed this sentence from the abstract. We have also extensively revised the manuscript to stress the descriptive nature of this study. Nevertheless, we believe that our observations may help to shed light on the dynamics of HIV-1 DNA profile changes during the earliest stages of infection.

Finally, the text really should be revised by someone who can do justice to the data by making the text easier to read with clearer and more correct prose. One example is the sentence in lines 185-187, but this sort of sentence detracts from the text throughout. Throughout, punctuation, abbreviations, grammar and word usage are sloppy. Also, usage of nouns as adjectives impairs the text, but I have elected not to detail these. The senior authors should put in more effort in revising the text.

Response: We apologize for the imperfect writing style of the first submission. We have now revised the text in this manuscript extensively in attempt to do justice to the data. Specifically, we have improved on sentence structures, punctuation marks usage, grammar, and word usage. We have also clarified all abbreviations, and have removed nouns that were used as adjectives.

REVIEWERS' COMMENTS:

Reviewer #1 (Remarks to the Author):

The authors have addressed some of the concerns raised in the initial review. In the abstract, they continue to frame the work in terms of viral reservoirs even though it remains unclear whether any of the viruses sequenced will become part of the stable reservoir. This should be rectified. In addition, they do not adequately discuss the most likely explanation for their findings, namely that in the setting of active viral replication, the sequences that will be the most prominent are those of the actively replicating virus (ie intact sequences).

Reviewer #2 (Remarks to the Author):

The authors have taken all of my suggestions into consideration and made a substantial effort to revise the manuscript. Much of my comments revolved around softening of the absolute and conclusive tone of the language and acknowledgement of limitations of the sample size as well as methodology. I commend the authors for acting on these suggestions, and I believe it has made this iteration of their manuscript more well-rounded and sound in its message. I recommend acceptance of this revised version.

Point-by-point Response to Reviewers' Comments:

REVIEWERS' COMMENTS:

Reviewer #1 (Remarks to the Author):

The authors have addressed some of the concerns raised in the initial review. In the abstract, they continue to frame the work in terms of viral reservoirs even though it remains unclear whether any of the viruses sequenced will become part of the stable reservoir. This should be rectified. In addition, they do not adequately discuss the most likely explanation for their findings, namely that in the setting of active viral replication, the sequences that will be the most prominent are those of the actively replicating virus (ie intact sequences).

Response: We have now removed all mentions of “viral reservoirs” in the abstract. We have also revised our concluding statement to stress that the HIV genomes detected in this study do not represent the persistent viral reservoir, and that a study with a longer sampling time frame is needed to understand viral and host factors that contribute to viral persistence. We do appreciate the reviewer’s focus on active viral replication as an explanation for the dominance of intact proviruses during acute infection, but would like to point out that active viral replication also occurs prior to treatment initiation in chronic infection. The striking discrepancy between the relative proportion of intact vs. defective proviruses in individuals with treatment initiation in acute vs. chronic infection is therefore unlikely to be solely related to active viral replication. We have edited the discussion to address this point.

Reviewer #2 (Remarks to the Author):

The authors have taken all of my suggestions into consideration and made a substantial effort to revise the manuscript. Much of my comments revolved around softening of the absolute and conclusive tone of the language and acknowledgement of limitations of the sample size as well as methodology. I commend the authors for acting on these suggestions, and I believe it has made this iteration of their manuscript more well-rounded and sound in its message. I recommend acceptance of this revised version.

Response: We thank Reviewer 2 for his/her encouraging comments.